# Intracellular signaling in proto-eukaryotes evolves to alleviate regulatory conflicts of endosymbiosis

**Samuel H. A. von der Dunk** \*, **Paulien Hogeweg, Berend Snel**

Department of Biology, Utrecht University, Utrecht, The Netherlands

\* samvonderdunk@hotmail.com

## Abstract

The complex eukaryotic cell resulted from a merger between simpler prokaryotic cells, yet the role of the mitochondrial endosymbiosis with respect to other eukaryotic innovations has remained under dispute. To investigate how the regulatory challenges associated with the endosymbiotic state impacted genome and network evolution during eukaryogenesis, we study a constructive computational model where two simple cells are forced into an obligate endosymbiosis. Across multiple *in silico* evolutionary replicates, we observe the emergence of different mechanisms for the coordination of host and symbiont cell cycles, stabilizing the endosymbiotic relationship. In most cases, coordination is implicit, without signaling between host and symbiont. Signaling only evolves when there is leakage of regulatory products between host and symbiont. In the fittest evolutionary replicate, the host has taken full control of the symbiont cell cycle through signaling, mimicking the regulatory dominance of the nucleus over the mitochondrion that evolved during eukaryogenesis.

## Author summary

Virtually all life forms visible by the naked eye are eukaryotes, complex cells with a nucleus, mitochondria and many other subcellular structures as well as the regulatory mechanisms to organize this intricate subcellular environment. Most steps in the complexification of eukaryotes still remain under dispute due to limited direct data. However, it is clear that eukaryotes originate from the merger between two simpler cells that were engaged in an endosymbiotic relationship, i.e. the mitochondrial endosymbiosis. We here use a multilevel computational model to investigate how this crucial event in the origin of eukaryotes could contribute to the evolution of complexity. As many details about the endosymbiosis are unknown, our model confronts host and symbiont organisms with several fundamental challenges that we hypothesize to arise from the nature of their relationship—such as coordination of growth and coping with transfer of regulatory molecules and DNA. To overcome these challenges the cells in our model evolve new regulatory mechanisms and various communication channels. Moreover, multiple replicate evolutionary trajectories lead to various alternative control strategies, allowing us to broadly

**Data Availability Statement:** Data were generated from a custom-build computational model written in C++. The annotated code of the model is

available at https://github.com/samvonderdunk/
Eukaryotes. Data, including full backups and
ancestry files are available on the FigShare of Sam
von der Dunk under the name "Supplementary
Data for Eukaryotes model with leakage and
transfer": https://figshare.com/articles/dataset/
Supplementary_Data_for_Eukaryotes_model_
with_leakage_and_transfer/24631023.

**Funding:** This work was supported by The
Netherlands Organisation for Scientific Research
(NWO-Vici 016.160.638 to B.S.): https://www.nwo.
nl/. The funder did not play any role in the study
design, data collection and analysis, decision to
publish, or preparation of the manuscript.

**Competing interests:** The authors have declared
that no competing interests exist.

explore the consequences of an obligate endosymbiotic relationship and its impact on genomic and regulatory complexity.

## Introduction

The mitochondrial endosymbiosis was an important step in eukaryogenesis, which gave eukaryotes a crucial energetic benefit and increased evolutionary potential [1–4]. Yet the impact of the mitochondrial endosymbiosis on eukaryogenesis reaches beyond increased metabolic capacity. Through endosymbiotic gene transfer, the large majority of mitochondrial genes ended up in the nuclear genome contributing to its expansion [5, 6]. In addition, the nucleus evolved regulatory control over mitochondria, subjugating them to the eukaryotic cell cycle (e.g. [7–9]). Several other eukaryotic innovations—including the nucleus, intron splicing and sex—have also been posited to be direct consequences or adaptations to the mitochondrial endosymbiosis [10–13], although the mitochondrial impetus for the evolution of the nucleus and introns has recently been called into question [14].

The mitochondrial endosymbiosis was unique relative to all known subsequent endosymbioses (including the chloroplast) as these involved a eukaryotic host which had already adapted to endosymbionts (i.e. mitochondria). These mitochondria-carrying eukaryotic hosts feature a well-defined subcellular organization as well as several distinct regulation and signaling mechanisms to coordinate endosymbiosis. Likely as a consequence, eukaryotes commonly engage in endosymbiosis of prokaryotes and of other eukaryotes [15–17]. Prokaryotes, on the other hand do not feature subcellular organization and frequently engage in *ecto*symbiotic but not in *endo*symbiotic relations (with one notable exception of $\gamma$-proteobacteria inside $\beta$-proteobacteria within the mealybug [18]). We hypothesize that an endosymbiosis between two prokaryote-like organisms presents many challenges, such as cell-cycle coordination of endosymbionts, and preservation of genomic and functional integrity of the host despite leakage of symbiont molecules. Identifying mechanisms by which evolution can solve these challenges is important for understanding eukaryotic life.

Modeling is uniquely suited to obtain insights into how evolution could potentially have overcome the challenges of endosymbiosis. As outlined, eukaryogenesis and endosymbiosis are complicated processes that involve multiple levels of organization, e.g. genomic, regulatory, cellular, holobiont. On top of that, eukaryogenesis occurred only once, roughly 1.7–2.4 billion years ago, leaving little direct data to probe the evolutionary forces that were at play [19, 20]. We recently developed a multilevel model based on cell-cycle regulation, which is well suited to investigate endosymbiosis in an evolutionary context [21]. The cell cycle is a fundamental task of a cell which connects many levels of organization: from specific regulatory products and binding sites on the genome to the overall behavior of the cell (growth, replication, division). Furthermore, the cell cycle is likely a focal point of evolutionary changes in an endosymbiotic context, where host and symbiont need to coordinate their growth and division cycles relative to each other. Using our multilevel model, we investigate how an endosymbiosis event like the mitochondrial endosymbiosis can impact cells at the level of genome, regulatory network, and cell-cycle dynamics.

## Materials and methods

### Cell-cycle regulation in host and symbiont

To model obligate endosymbiosis including molecular interference between host and symbiont, we use our previous model of obligate endosymbiosis and cell-cycle regulation ([22],

which is an extension of [21] and [23]), and extend it with product leakage, targeting and gene transfer (Fig 1). Hosts and symbionts are modeled as entities that regulate an autonomous cell cycle consisting of four states defined by the expression of five core gene types (g1–5): G1; the S-phase in which the genome has to be replicated; G2; and the M-stage in which cells divide. If a cell reaches M-stage too early, i.e. without having passed through all preceding stages or without having finished genome replication, it dies. The timely expression of cell-cycle stages is achieved by a Boolean gene regulatory network that is formed by the interactions between discrete regulatory genes (coding for regulatory products) and binding sites, encoded on a linear genome (a representation known as a beads-on-a-string genome; e.g. [24]). Genes become expressed when sufficiently many excitatory products are bound to their upstream binding

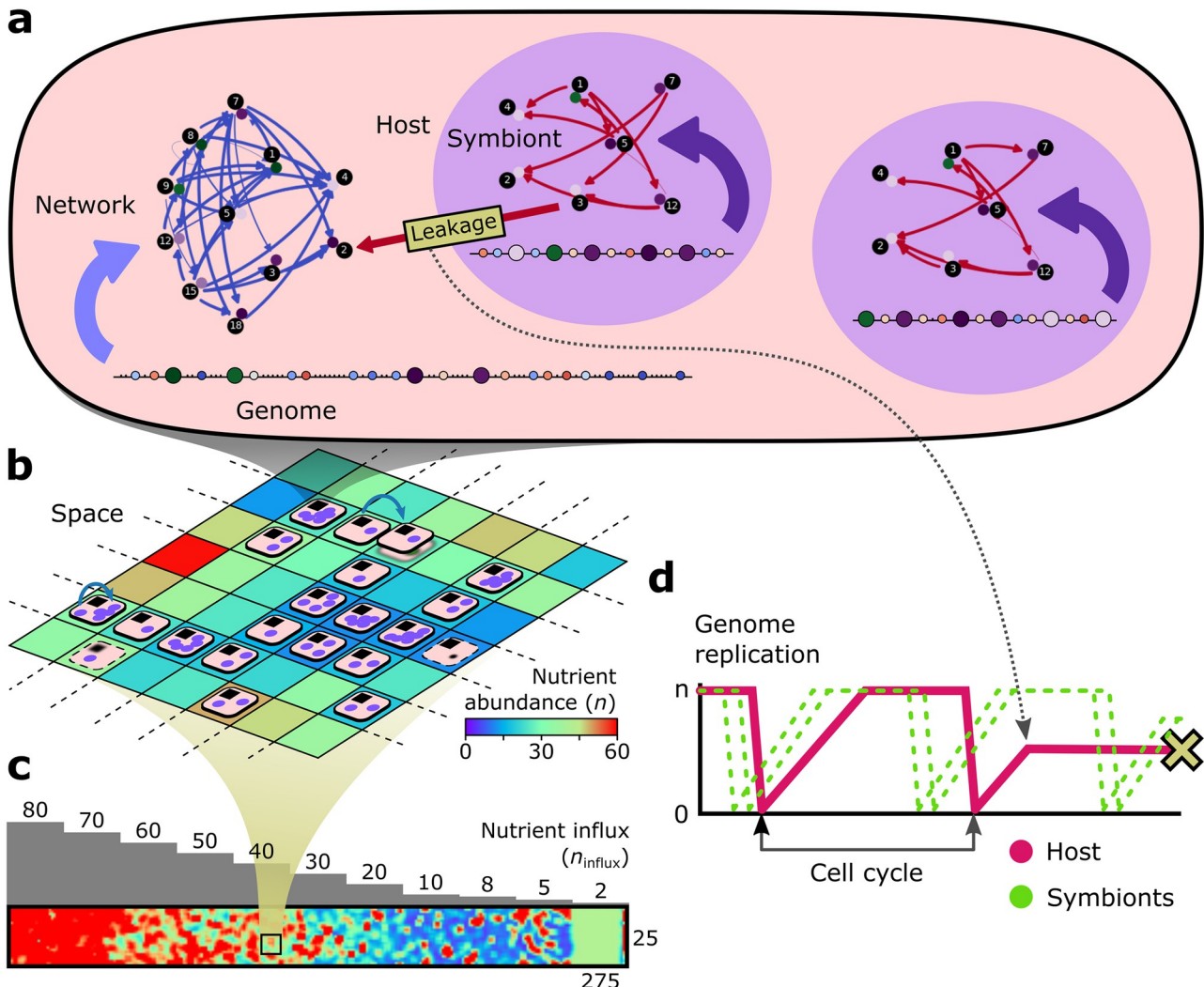

**Fig 1. Overview of the modelling framework (cf. [22]).** (a–c) Holobionts which consist of a host and one or more symbionts live on a grid where they compete for nutrients. (a) At the core of each host and symbiont is a genome with beads representing regulatory genes, binding sites and passive household genes. Interactions between gene products and binding sites give rise to the regulatory network and expression dynamics constituting the cell cycle (d). The host and symbionts interact through product leakage, product targeting (i.e. signaling) and gene transfer. As an example, leakage of symbiont g3 product to the host is shown and the impact on cell-cycle dynamics: (d) the host getting stuck mid-way through replication and later dying trying to divide with a partially replicated genome. In this and subsequent figures, regulatory interactions mediated by products that originate from the host are colored blue and those that originate from the symbiont are colored red.

**Table 1. Mutation rates.**

| Parameter | Value | Description |
|---|---|---|
| $\mu_B$ | $3.3 \cdot 10^{-5}$ | Per-bit mutation rate of binding sequence |
| $\mu_S$ | $1.0 \cdot 10^{-5}$ | Per-bit mutation rate of signal peptide |
| $\mu_w$ | $1.67 \cdot 10^{-4}$ | Regulatory weight mutation rate |
| $\mu_\theta$ | $1.67 \cdot 10^{-4}$ | Activation threshold mutation rate |
| $\mu_{dup}$ | $1.67 \cdot 10^{-4}$ | Per-bead duplication rate |
| $\mu_{del}$ | $1.67 \cdot 10^{-4}$ | Per-bead deletion rate |
| $\mu_{rel}$ | $1.67 \cdot 10^{-4}$ | Per-bead relocation rate |
| $\mu_t$ | $2.0 \cdot 10^{-5}$ | Per-bead transfer rate |
| $\mu_{b,in}$ | $1.67 \cdot 10^{-3}$ | Per-genome innovation rate for binding sites |
| $\mu_{r,in}$ | $1.67 \cdot 10^{-4}$ | Per-genome innovation rate for regulatory genes |

sites to surpass their specific activation threshold. An expressed gene has a probability to bind to any binding site on the genome which depends on the bitstring similarity between gene and binding site (see [21]). Per timestep, at most one product binds per binding site, resulting in stochasticity in the regulatory interactions: even pairs of gene and binding site with identical bitstrings do not bind 100% of the time. Moreover, cells can evolve low-affinity interactions which only occur rarely and which can be used to time certain cell-cycle stages, such as a long-lasting S-phase to adapt to slow replication in poor nutrient conditions (see below). Low-affinity interactions are also sensitive to the dosage changes that occur during genome replication (see below), allowing cells' regulatory networks to respond to replication progress. Through the evolution of a specific genome organization, these dosage changes can give rise to a *de novo* cell-cycle checkpoint that stalls the cell cycle and delays division until replication is finished [21]. The gene regulatory network evolves through mutations in sequences and other properties of genes and binding sites (regulatory effect, activation threshold), as well as through duplications, deletions, innovations and relocations of genes and binding sites on the genome (Table 1).

## Balancing symbiont number with nutrient availability

Obligate endosymbiosis is modeled as holobionts consisting of one host and one or more symbionts, which live on a two-dimensional grid with an externally provided nutrient gradient (Fig 1). Hosts and symbionts are modeled individually, allowing for the existence of holobionts with various unique symbionts. The holobiont dies when the host dies or when there are no symbionts left. The holobiont replicates into a new grid site when the host divides. Symbionts are stochastically distributed over offspring, so holobionts need to maintain more than one symbiont at the time of division to ensure that each daughter receives at least one symbiont. At the same time, hosts and symbionts share the nutrients in the environment (3-by-3 neighborhood), so higher symbiont number leads to lower nutrient availability. Nutrients are required for replication, defining how many genomic elements (regulatory genes, binding sites, and passive household genes) are replicated during one timestep in S-phase. Thus, to achieve fast growth, holobionts should not maintain too many symbionts.

## Extending the endosymbiosis model with molecular interference

To study the evolutionary challenges of endosymbiosis, we model interference between the gene regulatory networks and genomes of host and symbiont, representing the holobiont

before the emergence of structures or mechanisms that act as boundaries, such as the nucleus (Fig 1). Molecular interference is implemented in both directions as to remain agnostic to the specific mechanisms of interference, although symbiont death would make symbiont-to-host interference more likely than the reverse.

The products of expressed genes leak between symbionts and the host with a rate of $l = 0.01$ per gene per symbiont. As a consequence, the influx into each symbiont will be proportional to the number of expressed genes in the host, whereas the influx into the host will be proportional to the number of expressed genes in all symbionts combined. Foreign products act like native products in all respects: they can bind to binding sites and they also define the cell-cycle stage if they are identical to one of the native core gene types (g1–5). Thus, product leakage disrupts expression dynamics and hinders the initially autonomous cell-cycle regulation of hosts and symbionts (Fig 1).

Besides passive molecular interference, we include mutations that generate or destroy signal peptides of products ($\mu_S$, Table 1) yielding active targeting of products to the host or symbiont. The signal peptide for host and symbiont localization are represented by two bits, respectively: 10 defines host localization, 01 symbiont localization, 00 no relocation, and 11 dual localization. In the initial holobiont, all gene products are only targeted to the genome where they are encoded (10 or 01).

At the genome-level, interference consists in gene transfer between host and symbiont, which can disrupt the encoded gene regulatory network or foster innovation. A newly divided host (symbiont) can receive gene transfers from all symbionts (from the host) at a per-gene rate of $\mu_t$ (Table 1), and we included both copy-and-paste and cut-and-paste type transfers.

The multi-level model as described here is designed to explore evolution of an endosymbiotic system at various levels of biological organization. We find several unexpected outcomes which we subject to in-depth mechanistic and evolutionary analysis. In line with this approach, we do not perform large numbers of simulations or statistical analysis on our outcomes, as these are unlikely to reflect the specific outcomes of eukaryogenesis in nature for which we do not know all relevant system features let alone the initial conditions. Moreover, owing to the number of biological processes that are explicitly represented, model simulations are computationally demanding. When cells adapt, and populations and genomes expand, less than $4 \cdot 10^4$ timesteps can be simulated per day, leading some evolutionary experiments to last for more than six months.

## Results

### Holobionts adapt despite molecular interference

To study host and symbiont evolution in obligate endosymbiosis, we used our multilevel model of the cell cycle in the context of gene regulation. In short, genomes undergo DNA replication which has to be timed with cell division. Hosts and symbionts have to evolve their cell cycle to adapt to poor and fluctuating nutrient conditions on the external gradient which requires prolonged duration of the S-phase. At the same time, the holobiont must also evolve coordination of symbiont division to avoid losing the obligate symbiont. Because there are several organizational levels in our model and no explicit fitness criterion, evolution has many degrees of freedom to overcome the set challenges.

We evolved 10 replicate populations (Q1–10) with these endosymbiotic challenges, i.e. product leakage, gene transfer and signal peptide mutations. Each population is inoculated on a nutrient gradient (Fig 1c), where individuals are left to execute their cell-cycle behavior and evolve for $10^7$ timesteps. All replicates start with the same primitive host and symbiont genomes ($L = 64$ which includes 50 household genes), derived from [23] (see also [21, 22]).

Despite initial identical genomes and gene regulatory networks, host and symbiont cell cycles do not stay synchronized due to stochasticity in the regulatory dynamics. During the evolution experiment, progress is tracked in terms of population size (a proxy for success), symbiont number, and genome size of host and symbiont (a measure of complexity). After the evolution experiment, the ancestry of the final population at $t = 10^7$ is reconstructed, and the most recent common ancestor of each replicate is analyzed in detail using various additional experiments.

Adapting to the endosymbiotic lifestyle with the implicated challenges is difficult, as we describe in detail for the most successful replicate as measured by population size, Q4 (Fig 2; see Figs D–K in S1 Appendix for alternative trajectories). Early in the experiment, the population is small and confined to the richest environments on the gradient. A small population is

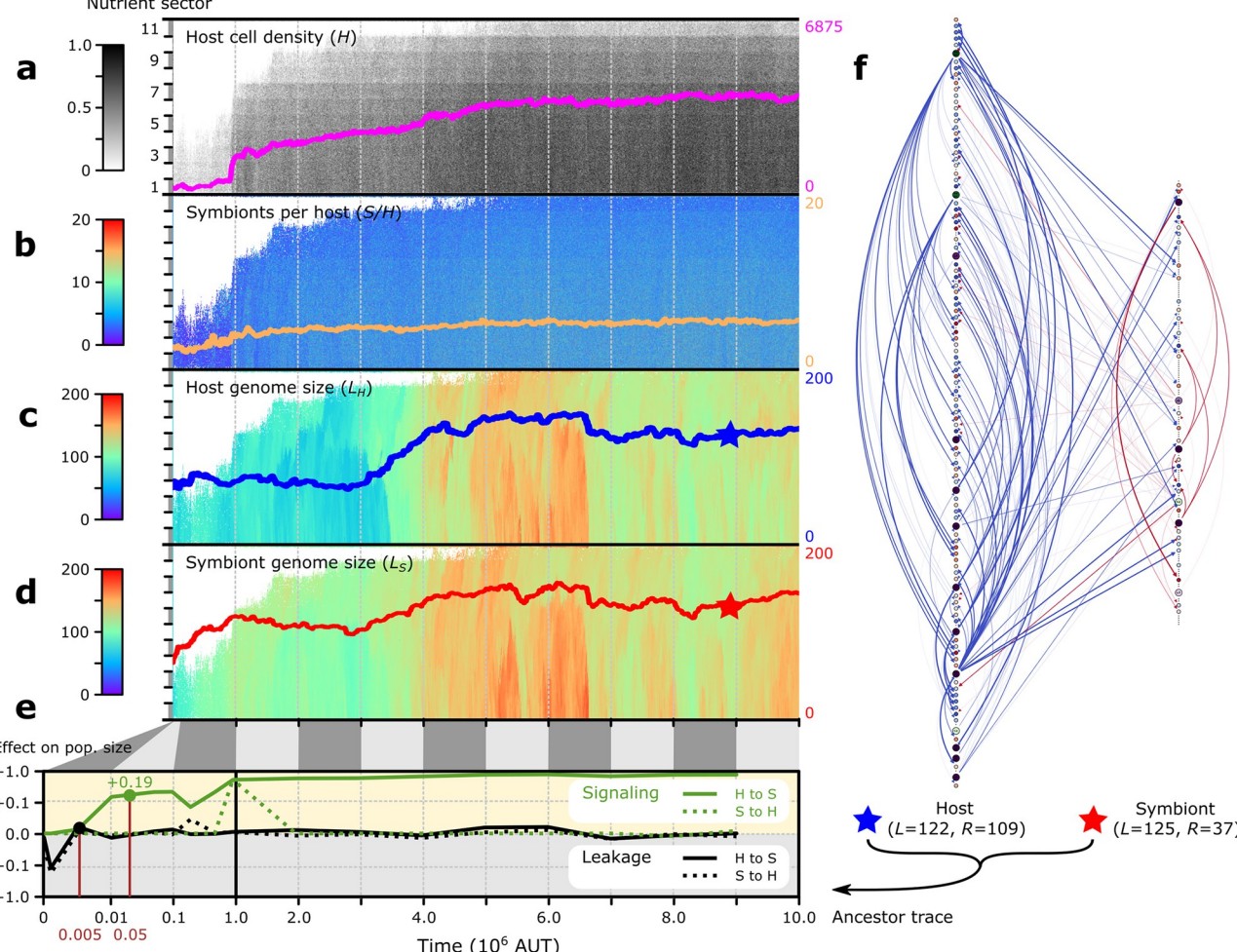

**Fig 2. Evolution of host–symbiont signaling in holobionts adapting to a nutrient gradient (replicate Q4; see Figs D–K in S1 Appendix for evolutionary dynamics in other replicates).** The four top panels (a–d) show the evolution of (a) population size, (b) symbiont number and (c,d) genome size of host and symbiont in space (the gradient with 11 nutrient sectors shown in the vertical dimension) and time (in the horizontal dimension). Overlaid on each space-time plot are the population averages across space. (f) The genome networks are shown of the last common ancestors (marked with a star in c,d) of all hosts and symbionts in the final population. The symbiont genome appears smaller than the host genome because it encodes mostly passive household genes (which are represented by small squares). Below the genomes, the total genome size ($L$) and size of the regulatory repertoire ($R$) are indicated. (e) We assessed the impact of leakage and signaling along the ancestral lineage at high nutrient conditions ($n_{influx} = 90$). For instance, the ancestor at $t = 0.05 \cdot 10^6$ forms a population size (grid density) of $N = 0.19$ under default conditions, but subsides ($N = 0$) when host-to-symbiont signaling is disabled; thus, host-to-symbiont signaling has a positive effect of + 0.19 at this time. The first $10^6$ AUT (arbitrary units of time, or timesteps) are shown on a semi-log scale, revealing very rapid evolution of signaling and insensitivity to leakage.

prone to extinction, as seen in 3 other replicates (Q1, Q2 and Q6; Fig 3). Moreover, selective forces are weak when the population is small and spread out over the grid, resulting in extensive drift of genome sizes: in the first $10^6$ timesteps of Q4, the symbiont genome expands to 186% of its initial size (i.e. from $L = 64$ to $L = 119.3$). It takes until around $t = 10^6$ for holobionts to become better adapted, as seen by rapid expansion of the population into poorer nutrient conditions on the gradient. At this time point, host and symbiont genome sizes no longer drift and decrease slightly showing that there is now stronger selection for shorter genomes that need less time for replication and thus allow execution of a faster cell cycle. Subsequently and concomitant with additional population growth, both host and symbiont genomes expand again between $t = 3 \cdot 10^6$ and $t = 5 \cdot 10^6$ and reach a similar size. Surprisingly, the symmetry in genome size is retained for the rest of the experiment, with both genomes simultaneously

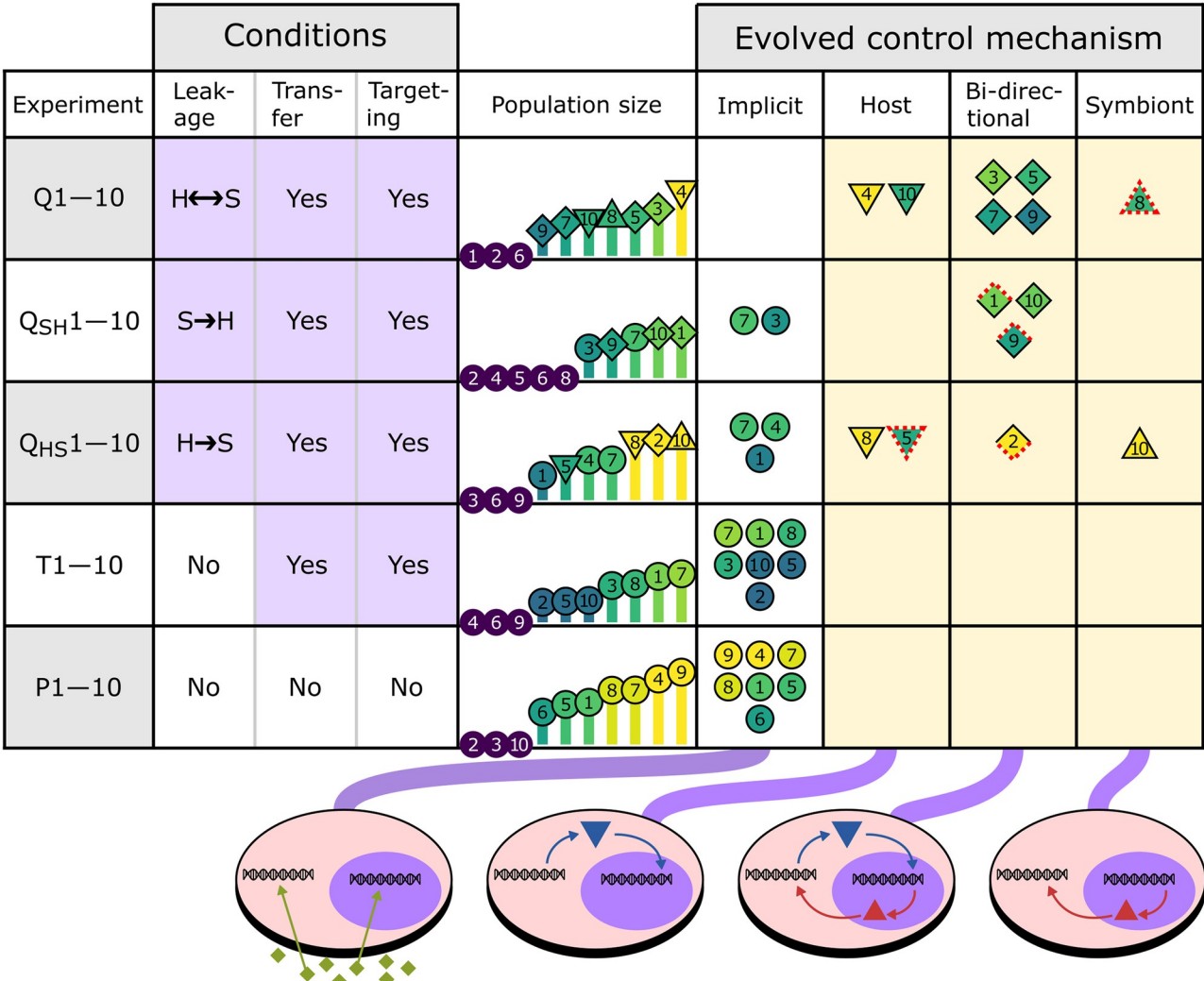

Fig 3. **Leakage drives evolution of signaling, as shown by outcomes of five evolution experiments performed with different leakage and transfer conditions.** With bi-directional leakage (first row), signaling always evolves (creme shading; see cartoons on the bottom). With uni-directional leakage (second and third rows), signaling evolves in some replicates. Without leakage (fourth and fifth rows), only implicit control evolves, i.e. cell-cycle coordination without regulatory communication. Triangles indicate the direction of control (down for host control, up for symbiont control, diamond for bi-directional control), and red dotted lines indicate communication through leakage (down for host-to-symbiont direction, up for symbiont-to-host direction).

shrinking or expanding at various occasions. Still, inspection of host and symbiont genomes reveals that they evolved encoding of different functions: the host has a large regulatory repertoire ($R$) whereas the symbiont is highly reduced in terms of regulation (Fig 2). Vice versa, the symbiont in Q4 encodes most of the passive household genes ($L - R$) for the holobiont, indicating that there is specialization of regulation and other (non-regulatory) functions between host and symbiont.

## Evolution of host–symbiont communication

As we set out to study the effects of molecular interference during endosymbiosis, we next investigated how holobionts adapted to product leakage along the ancestral lineage, as here detailed for replicate Q4 (Fig 2). To this end, clonal growth experiments were performed under high nutrient conditions ($n_{influx}$ = 90) with bi-directional, uni-directional or no product leakage between host and symbiont. A clonal population consists of a single holobiont type in which all hosts are genetically identical and all symbiont are genetically identical. For the first few ancestors, the carrying capacity is smaller when the clone is grown in the presence of leakage than when it is grown in the absence of leakage, confirming that product leakage is deleterious for early holobionts (Fig 2e). Yet Q4 holobionts have rapidly evolved to be largely insensitive to leakage: from $t = 0.005 \cdot 10^6$ onwards, there is almost no effect of leakage on population size (Fig 2e).

In parallel with the leakage assay, we also investigated whether host–symbiont signaling evolved along the ancestral lineage of replicate Q4. Studying genomes along the ancestral lineage, we find that genes occasionally obtain signal peptides through mutations such that the gene product is targeted from host to symbiont or from symbiont to host. However, many of the genes where this happens are not expressed and their cross-genomic interactions are not conserved through evolution. To detect whether *functional* signaling evolved, we studied how holobiont growth changes when targeting in one or both directions is blocked. Surprisingly, in Q4, host-to-symbiont signaling emerges very early and has already become essential for the holobiont by $t = 0.05 \cdot 10^6$ (Fig 2e). Functional signaling in the opposite direction, i.e. from symbiont to host, appears transiently at $t = 10^6$ but subsequently does not affect holobiont growth anymore. Thus, in Q4, passive communication in the form of product leakage is rapidly supplanted by active communication in the form of host–symbiont signaling (Fig 2e). Signaling by the host establishes regulatory control over all six symbiont regulatory genes, allowing the host to order symbiont division and effectively manage symbiont number. We refer to this particular evolved strategy as host control (see section "Host control achieves cell-cycle synchronization").

## Product leakage drives evolution of signaling

The 7 evolution replicates that survived and adapted to the nutrient gradient all evolved some form of signaling between host and symbiont (Fig 3). Besides host control as in Q4 (and also in Q10), we found symbiont control mediated by symbiont-to-host signaling (Q8), and bi-directional control mediated by two-way signaling (Q3, Q5, Q7 and Q9; see Figs L–P in S1 Appendix). To study whether product leakage or gene transfer played a role in the evolution of signaling, we performed several additional evolution experiments: with uni-directional leakage ($Q_{HS}1$–10 and $Q_{SH}1$–10), and without leakage altogether (T1–10). We also included in this comparison an earlier experiment without any interference or possibility for communication, i.e. no leakage, no transfer and no signal peptide mutations (P1–10; featured in [22]). Strikingly, signaling does not evolve in the absence of product leakage, despite mutations that generate signal peptides (T1–10). Moreover, under uni-directional leakage ($Q_{HS}1$–10 and $Q_{SH}1$–

10), signaling evolves in some but not all replicates. These observations firmly establish that product leakage drives the evolution of host–symbiont signaling in our model.

## Competitive success of evolved holobionts

To investigate how our evolution can give rise to such a diversity of control mechanisms across replicates, we first compare the fitness of holobionts between replicates. With signaling, Q4 holobionts achieve high stability and large population size. Yet, holobionts in P9 and P4 also reach large population size with a strategy to balance host and symbiont growth, despite the fact that no signaling could evolve in these replicates. Population size reflects the fitness of individual holobionts ($R_0$), but competitive success also depends on their generation time (i.e. at what timescale $R_0$ is realized) and interaction with the environment (i.e. growth depends on nutrient levels which in turn depend on population size and symbiont numbers). To compare evolved holobionts and their strategies directly, we performed various competition experiments between populations (Figs Q and R in S1 Appendix). Here, two final populations from different replicates are inoculated side by side and left to compete for $10^5$ timesteps or until one has taken over the entire grid. It turns out that holobionts from Q4 outcompete all other holobionts, because Q4 holobionts perform both accurate (high $R_0$) and fast cell cycles (short generation time). Nevertheless, other holobiont populations that evolved signaling (Q3,5,7–10 and $Q_{HS}$2,10) are outcompeted by P4 and P9, showing that signaling is not required for competitive success.

One important factor for competitive success of holobionts is the efficiency of individual host and symbiont cell cycles. At the start of the evolution experiment, host and symbiont regulate a primitive cell cycle. It takes substantial rewiring of the gene regulatory network to evolve a long and efficient cell cycle that can cope with poorer environments on the nutrient gradient [21]. In different replicates, hosts and symbionts achieve different levels of adaptive rewiring, and this translates to different levels of success in holobionts. For instance, host and symbiont cell cycles are more efficient in P9 than in Q3, as calculated by comparing the actual cell-cycle duration with the minimal cell-cycle duration for a given genome size ($e = \frac{\tau_{min}}{\tau}$, see Fig 4), and so P9 is more successful in competition. Thus we find that with many degrees of freedom, evolution can take various alternative paths, which can lead to different mechanistic solutions for a given challenge and can also lead to differences in adaptive success.

## Mechanisms of cell-cycle coordination

Our modeling approach provides us with the opportunity to investigate alternative outcomes of endosymbiosis and therewith contextualize the single outcome of eukaryogenesis that appeared 1.7–2.4 billion years ago [19]. Furthermore, our modeling approach allows us to pinpoint how different signaling mechanisms and control strategies work. To this end, we shall describe in the following sections for each holobiont strategy one successful replicate in terms of genome and network organization, regulatory behavior and cell-cycle dynamics (Fig 4). Cell-cycle dynamics were distilled during a clonal growth experiment at intermediate nutrient condition ($n_{influx} = 30$) without product leakage. We followed a holobiont in a specific site on the grid, studying changes in symbiont number and in cell-cycle regulation of symbionts and the host through several holobiont divisions.

## Host control achieves cell-cycle synchronization

As discussed above, holobionts in the most successful replicate out of all experiments (Q4) evolved host control, which seems to capture eukaryogenesis to the extent that much of mitochondrial biology is under regulatory control of the nucleus in extant eukaryotes [7–9]. Early

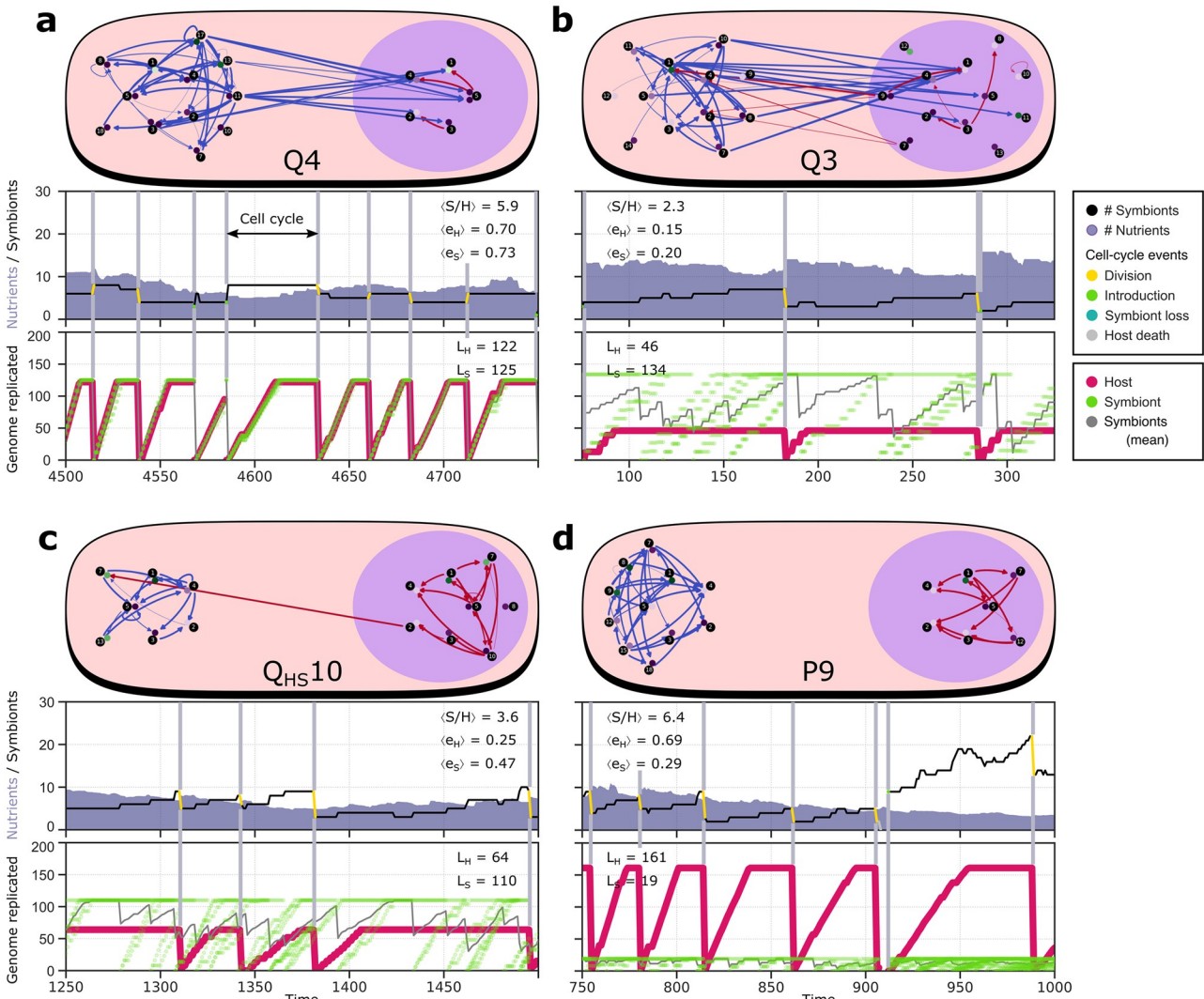

**Fig 4. Four unique evolved mechanisms of cell-cycle coordination: (a) host control, (b) bi-directional control, (c) symbiont control, and (d) implicit control.** For each mechanism, the most recent common ancestor of a successful replicate is analyzed in detail. The gene regulatory network of the holobiont is shown at the top, with regulation by host genes in blue and regulation by symbiont genes in red. Interactions show the aggregate regulatory effect (binding probability times regulatory weight) of gene types (black disks) on loci (colored disks adjacent to the gene types), of which there can be multiple with the same type (note e.g. two copies of g5 in the symbiont of Q4). In the two panels below, the cell-cycle behavior is shown, which was analyzed in clonal growth experiments under intermediate nutrient conditions ($n_{influx}$ = 30). The middle panel shows the number of symbionts in a holobiont (black lines) during several division cycles (grey vertical lines); the bottom panel shows the individual cell cycle progress of host and symbiont, as measured by how much of the genome has been replicated (genome size can be read off from the maxima). From the entire time data ($10^4$ timesteps), the average symbiont number at birth ($S/H$) and regulatory efficiencies of host and symbiont ($e_H$ and $e_S$) were also calculated (see main text; [22]). High cell-cycle efficiency of the host, as seen in smooth replication trajectories (as opposed to more step-like), is partly responsible for the high fitness of Q4 and P9.

on in Q4, several host products (starting with products of core genes g1 and g4) evolve dual localization, promoting the same expression dynamics in the symbiont as in the host (Fig S in S1 Appendix). Over time, the symbiont copies of g1 and g4 lose their regulatory functions and these are taken over by the versions imported from the host. At the end of the experiment, only three important regulatory interactions are still carried out by the symbiont itself (Fig 4a). The remaining symbiont genes (g1–5) are still functional as they define the symbiont's cell-

cycle stage. These symbiont core genes have diverged from the core genes of the host, which prevents the latter from taking over and defining cell-cycle expression for the symbiont (Table C in S1 Appendix).

Host control results in synchronization of host and symbiont cell cycles (Fig 4a), something that is often considered an early and important step in the evolution of endosymbiosis [25]. Synchronization ensures that each symbiont divides once for every host division, resulting in constant symbiont number during the lifespan of a holobiont. Because each symbiont divides once, new holobionts on average carry the same number of symbionts as their parents. Stochastic differences in symbiont number, which arise from random distribution of symbionts at holobiont birth, are exclusively corrected at the holobiont level (see Text B in S1 Appendix). Holobionts with too few symbionts do not divide successfully and holobionts with too many symbionts grow slow due to low nutrient levels; both are outcompeted by holobionts that happened to receive an intermediate number of symbionts at birth.

Host control creates selection for equal genome sizes (as observed in Fig 2). The holobiont life cycle is most efficient when replication of host and symbiont genomes takes the same amount of time, since division needs to be delayed by the host's regulatory network until both genomes are fully replicated. Thus, cell-cycle synchronization promotes symmetry in genome size despite functional differentiation as observed before (Fig 2).

## Bi-directional control tunes symbiont number

The most common signaling mechanism in our evolution experiment is bi-directional control (8 replicates; Fig 3). The initial evolution of bi-directional control is similar to that of host control: host products obtain dual localization early in the experiment and start exerting regulatory control over symbiont gene expression (Fig T in S1 Appendix). Subsequently, a symbiont product with host localization appears and fixes in the population. For example in Q3, the gene g9 is encoded on the symbiont and its product targeted to the host where it triggers cell-cycle progression towards M-stage, i.e. promoting division (Fig 4b). In turn, g9 expression relies on activation from the symbiont-targeted host product g10 which binds the regulatory region upstream of g9 with low binding affinity. Interactions with low binding affinity are sensitive to gene dosage, meaning that replication of the host genome (increasing copy number of g10) and replication of symbionts (increasing copy number of g9) both increase the probability of host division. Moreover, g10 is located near the terminus of the host genome. Thus in Q3, the host has evolved a checkpoint that stalls the cell cycle until the host has finished genome replication and until there are enough symbionts to safely divide the holobiont.

Bi-directional control has exposed a different mechanism for control of symbiont numbers relative to synchronization under host control (Fig 4b). Holobionts with bi-directional control display complex developmental behavior that resembles a real life cycle: during the lifespan of a holobiont, symbiont number gradually increases resulting in further depletion of nutrients and slowing down of replication. When a holobiont is "old" and carries many symbionts, it divides and gives birth to two "young" holobionts carrying on average half the number of symbionts. Importantly, deviations from this average are corrected in the next cell cycle through the aforementioned division checkpoint. Thus, symbiont numbers are not controlled at the holobiont population level as in the case of host control, but at the level of individual regulation, i.e. through signaling between host and symbiont.

Interestingly, in holobionts with bi-directional control, the host drives the symbiont cell cycle to a large extent: in Q3, the host regulates 6 symbiont genes whereas the symbiont only regulates a single host gene (Fig D.f in S1 Appendix). Thus, host and symbiont have both lost

autonomy, but the host is more dominant in terms of regulation reminiscent of real endosymbiotic relationships.

## Symbiont control

The least common signaling mechanism that evolved is symbiont control (2 replicates; Fig 3), and $Q_{HS}10$ is the most successful replicate with this mechanism (Fig 4c). In contrast to host-to-symbiont signaling (e.g. in Q3 and Q4, see Fig 2e and Fig D.e in S1 Appendix), symbiont-to-host signaling arises very late in evolution (e.g. in Q3 and $Q_{HS}10$, see Figs D.e, J.e and K.e in S1 Appendix). In $Q_{HS}10$, the symbiont gene product g2 evolves dual localization around $t = 3 \cdot 10^6$, but only establishes functionally relevant regulation of the host around $t = 9 \cdot 10^6$ (Fig U in S1 Appendix). Specifically, g2 targets and inhibits host g7, the gene product that stalls the host cell cycle in S-phase and delays division. Symbiont g2 is expressed infrequently during the symbiont cell cycle and rarely at the time that is required to promote host division. Higher symbiont number increases the probability that a g2 copy from any symbiont is expressed at the right time. Thus, similar to bi-directional control, symbiont control acts to inform the host that there are enough symbionts to safely divide the holobiont. Interestingly, hosts in $Q_{HS}10$, unlike those in Q3, use two independent checkpoints to check their own replication status and the number of symbionts before dividing (see Text C in S1 Appendix).

## Product leakage exploited as additional channel for communication

Interestingly, we also found cases of host control, bi-directional control and symbiont control that rely on product leakage rather than signaling (see Fig 3). For instance in Q8 (an outcome with symbiont control), we determined from clonal growth experiments that hosts occasionally enter a dormant state in their cell cycle where no genes are expressed. In this state, they depend on the leakage of symbiont products to start up their cell cycle again. As the rate of product leakage scales with symbiont number, symbiont control through passive leakage and symbiont control through signaling both work by stalling the host cell cycle until there are enough symbionts to safely divide the holobiont.

In $Q_{HS}2$ (a case of bi-directional control), the host leaks the product of gene g1 to the symbiont, where it activates gene g14 whose product is then actively targeted back to the host to induce cell-cycle completion. Thus, product leakage and signaling can operate in unison to achieve cell-cycle coordination. In the same way that low binding affinity sets a timescale on a regulatory interaction, the leakage rate also sets a timescale on regulatory events. Moreover, low binding affinity interactions and effective leakage rates are both sensitive to gene dosage (i.e. expression frequency, replication status, symbiont number) and can thus be used as cell-cycle checkpoints that integrate external information into the regulatory network. In this light, it is interesting to note that $Q_{HS}2$ is much more successful than Q3 which uses signaling in both directions for bi-directional control (Fig 3).

We have seen that signaling does not evolve without leakage and that leakage itself can also be exploited for host–symbiont coordination. These observations could suggest that such functional leakage acts as an intermediate step in the evolution of signaling, whereby a host (symbiont) product first acquires a regulatory role in the symbiont (host) mediated by leakage and then later acquires the target signal. This is not the case: as we saw in Q4 (Fig 2e), leakage does not perform a functional role in the holobiont prior to the emergence of signaling. It appears that the critical effect of leakage is that it compromises host and symbiont cell-cycle autonomy, which is vital for execution of the distinct host and symbiont cell cycles that accomplish implicit control (see below). Leakage thereby stimulates the integration of regulatory networks

through communication to overcome its negative effect on implicit control, and this integration is achieved through active signaling or by exploiting leakage directly.

## Implicit cell-cycle control indirectly tunes symbiont number

The most common evolved coordination strategy—and the only outcome in the absence of product leakage—is implicit control, which we discovered as outcome in our previous study (Fig 4d; [22]). In this strategy, the host does not explicitly communicate with symbionts to decide when it divides. Instead, hosts and symbionts have evolved specialization on distinct nutrient conditions and form a stable equilibrium in their growth dynamics. Fluctuations in the local nutrient condition (indirectly reflecting changes in symbiont number) are countered by an increase or decrease in the growth rate of symbionts relative to the host until the system returns to the equilibrium state where host and symbiont grow at the same rate. It is remarkable that autonomous cell-cycle behavior of host and symbiont achieves such successful coordination, especially since nutrients are averaged over the local environment and thus only provide a coarse measure of symbiont number inside the holobiont. Moreover, in comparison to bi-directional control and symbiont control, implicit control takes several holobiont generations to correct high or low symbiont numbers resulting from asymmetric holobiont division.

In most replicates with implicit control, there is symmetry breaking in genome size between the host and the symbiont, as detailed in our previous study [22]. This can be explained by selection on the relative cell-cycle speeds of host and symbiont. In P9 for instance, to achieve a stable holobiont with high symbiont number, the host evolves a slower cell cycle than the symbiont. With a slower cell cycle, there is less selection for genomic streamlining and the host evolves to take over household functions from the symbiont—which regulates a faster cell cycle and accordingly experiences more selection for genomic streamlining. Not all replicates evolved large host and small symbiont genomes, but those replicates that did performed best in competition experiments (e.g. P9 and P4; see also Figs Q and R in S1 Appendix).

## Historical contingency favors implicit control

All the experiments described so far involved identical and primitive initial host and symbiont genomes. In the light of eukaryogenesis, it is interesting to also consider the effects of molecular interference between a host and symbiont that are more distantly related and which are already well adapted to the nutrient gradient. For this, we exposed the 7 surviving replicate populations that evolved without leakage and transfer and signal peptide mutations (P1–10) to all of these processes from $t = 10^7$ onwards (Fig 5). The 7 replicate populations had all evolved implicit control (Fig 3). Three replicates evolved high symbiont number resulting in stable, slow-growing holobionts and four replicates evolved low symbiont number resulting in unstable, fast-growing holobionts.

The introduction of molecular interference disrupts holobiont behavior resulting in immediate population size reductions (by up to 66.7% in the first 1000 timesteps; Fig 5). Thus, despite pre-adaptation of host and symbiont to the obligate endosymbiosis, molecular interference still presents a hazard for correct regulation of their cell cycles. Product leakage has the biggest direct impact on holobiont growth, although gene transfer also impacts holobionts on a long timescale by altering genome size evolution (see Text A in S1 Appendix). Replicates that had evolved high symbiont number before the transition to the new regime are impacted more than those that had evolved low symbiont number, because effective leakage rates are higher in holobionts with more symbionts. Holobionts quickly adapt to product leakage by decreasing symbiont number. Yet, the mechanism of cell-cycle coordination does not change. The genetic distance between host and symbiont makes signaling difficult to evolve in our genotype-

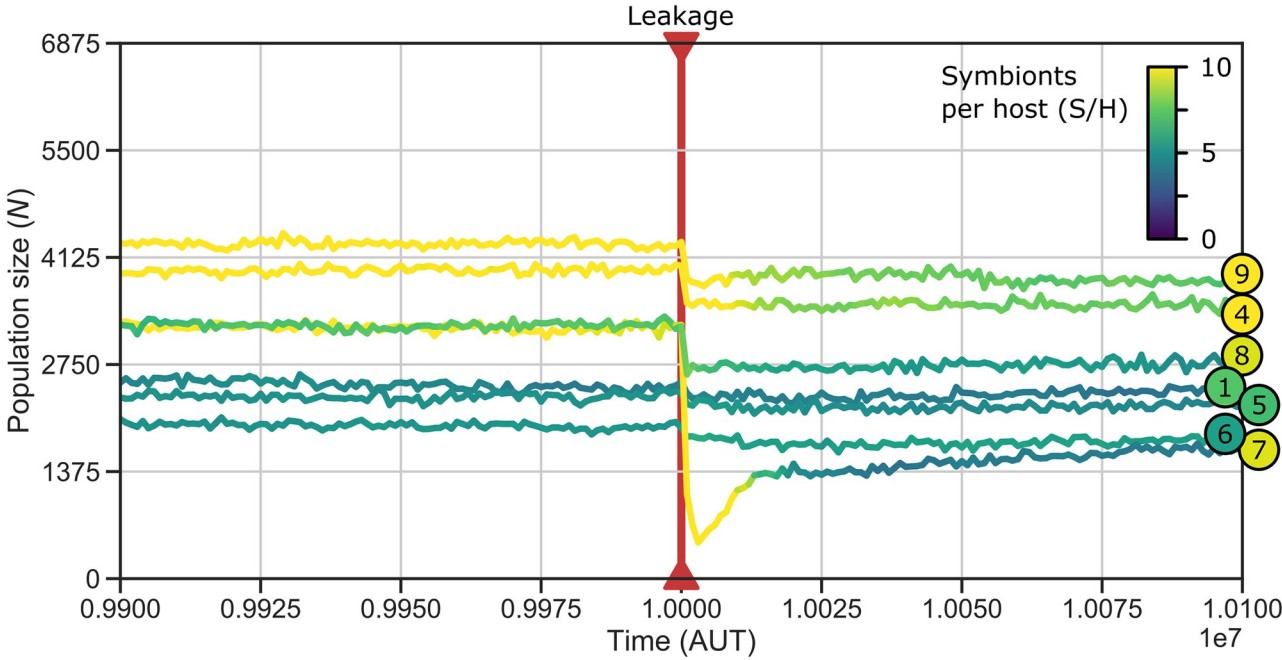

**Fig 5. Product leakage hinders holobionts even after they are adapted to the endosymbiotic state and host and symbiont have diverged.** The 7 surviving replicates of P1–10 (badges on the right side match those in Fig 3) were exposed to leakage, transfer and signal peptide mutations from $t = 10^7$, causing immediate drops in population size. Populations which evolved high symbiont numbers are affected most: P7 nearly goes extinct. Holobionts readapt by decreasing symbiont number.

phenotype map. Implicit cell-cycle control does not require genetic integration and is therefore much easier to evolve between distantly related host and symbionts. The same outcome was also clearly visible in another set of experiments starting with pre-evolved free-living host and symbionts (see Text A in S1 Appendix). Thus, historical contingency, i.e. the evolutionary past of free-living cells before they engage in endosymbiosis, favors a fully autonomous host and symbiont which balance their nutrient-dependent growth rates.

## Discussion

### Evolution of signaling and regulatory dominance of the host

We investigated the impact of obligate endosymbiosis in simple cells that are not pre-adapted to the prevailing environment and in particular have never been exposed to the physical challenges of endosymbiosis. The main challenge of endosymbiosis is to control the symbiont population through cell-cycle coordination between host and symbiont. Additionally, product leakage disrupts the regulatory autonomy of host and symbiont. Here we show that product leakage is not only a challenge but in fact also drives the evolution of intracellular signaling, indirectly aiding the establishment of cell-cycle coordination and stable holobiont growth.

We have explored different endosymbiotic challenges and uncovered different signaling mechanisms that enable host and symbiont to stabilize the endosymbiotic relationship. In the replicate that performed best across all *de novo* evolutionary replicates, holobionts evolved host control, where the host controls the cell cycle of symbionts. Host control has also been the outcome of eukaryogenesis and all known secondary endosymbiosis events. In the more recent endosymbiosis of *Paulinella chromatophora* and its chromatophores, we already see signs of

host control, by loss of symbiont genes or transfer to the host genome [26]. Our model shows that alternative mechanisms and coordination strategies can arise and that these can also be very successful. Thus, it remains an open question why host control evolved during eukaryogenesis.

An interesting analogy can be made between control of symbiont numbers and cell size control, an actively studied research topic within cellular biology (see Text B in S1 Appendix [27–29]). Both prokaryotic and eukaryotic cells display size control, whereby stochastic differences in cell volume disappear with time. So far, single-cell studies have found that various bacterial species behave as adders: they add a fixed volume per cell cycle which dilutes differences in cell size over multiple generations [27, 29]. Instead, fission yeast is found to behave as a sizer: it divides when its cell volume reaches a specific target volume [27, 30]. The bi-directional control and symbiont control strategies are analogous to the sizer mechanism, since they link host division to symbiont number and therefore swiftly correct fluctuations in symbiont number arising at the beginning of the cell cycle (i.e. due to random distribution of symbionts at host birth). Interestingly, *Caulobacter crescentus* on which our cell-cycle model is based, behaves as a timer: it executes a cell cycle of a fixed duration such that there is no control of cell size [27, 31]. Because cell volume grows exponentially, large cells become even larger and small cells remain relatively smaller. The host control strategy resembles this behavior, because the host ensures division of all its symbionts during a cycle: holobionts with many symbionts make offspring with many symbionts, and holobionts with few symbionts make offspring with few symbionts. As mentioned, symbiont numbers are kept in balance by selection at the holobiont level, which implicates a high death rate of holobionts and which makes it all the more surprising that the host control strategy is so successful in our evolution experiment (see Text B in S1 Appendix).

In all evolved control mechanisms, the host turns out to be dominant in terms of regulation. In replicates with bi-directional control, hosts regulate several symbiont genes whereas symbionts regulate only one or two host genes. Among replicates with symbiont control, a single symbiont product is targeted to the host in one replicate ($Q_{HS}10$), and only leakage from symbiont to host is exploited in the other replicate (Q8). These observations invoke the image of symbionts merely informing the host rather than completely controlling it. The host could potentially still re-adapt to a free-living lifestyle by removing its single dependency on regulation by the symbiont, whereas the symbiont has become entrenched in the host (except in symbiont control).

At the level of genome size, no general asymmetry between host and symbiont was found to evolve, unlike in our previous model [22]. By extending the model with product leakage and transport, we encountered new ways for stabilization of endosymbiosis that make use of explicit regulatory communication and which come with different constraints on host and symbiont genomes compared to implicit control. In our modeling formalism, cell-cycle speed poses a major constraint on genome size, and to limit symbiont numbers and regulatory conflicts, symbionts evolve relatively slow cell cycles. Having slower symbiont than host cell cycles results in larger symbiont than host genomes in many replicates, which does not match the outcome of eukaryogenesis. However, genome size asymmetry has classically been explained from energetic [2] or mutational perspectives [32, 33] rather than a purely informational one. These other perspectives could provide potential additional reasons for the evolution of genome size asymmetry between host and symbiont.

## Mechanistic insights into the evolution of signaling

Signaling is often assumed to be readily evolvable as an adaptation to obligate endosymbiosis. Yet in our model, signaling only evolves under stress from product leakage and with simple

and identical initial host and symbiont genomes. The integration of gene regulatory networks becomes harder when host and symbiont genomes are very different or already execute efficient autonomous cell cycles before being forced into an endosymbiotic relationship. In these cases, implicit control evolves rapidly because it only requires fine-tuning of the existing regulatory behavior to particular nutrient conditions. Interestingly, we had expected gene transfer to enable the evolution of signaling, but gene transfers are mostly deleterious and rarely fixed by selection. Signaling emerges by relocalization of existing products to the endosymbiotic partner. This indicates that is easier for a gene whose expression is already controlled by the cell cycle to evolve a new function (i.e. neo-functionalization of the gene product) than for a gene with a new function to evolve to be under proper cell-cycle control (i.e. neo-functionalization of the regulatory region). This is likely due to the fact that in our model, regulatory regions are relatively large (spanning several binding sites), such that adaptation of the regulatory region requires more mutations than adaptation of the gene itself.

Using a multilevel computational model and *in silico* evolution experiments, we have shown that cell-cycle coordination in holobionts can be achieved through implicit or through explicit control. Implicit control occurs at the metabolic level, i.e. through interaction of host and symbiont with nutrients in the external environment. Explicit control occurs at the gene regulatory network level, i.e. through signaling or leakage, which allows for more direct and tight control of symbiont numbers. In both cases, the regulatory repertoire of the host is generally dominant over the symbiont, even if it the host cell cycle is under regulatory control of the symbiont. The general picture that emerges is that an endosymbiotic relationship can be stabilized at multiple organizational levels, and even in case of harsh conflicts at the molecular level. From an informational viewpoint, stable coordination of endosymbiosis does not present a hurdle for simple cells (prokaryotes); rather, endosymbiotic challenges help to explain how regulatory complexity increased during eukaryogenesis, and in particular, the molecular conflicts between host and symbiont may explain the emergence of intracellular signaling in proto-eukaryotes.

## Supporting information

**S1 Appendix. Supplementary materials for intracellular signalingin proto-eukaryotes evolves to alleviate regulatoryconflicts of endosymbiosis.**
(PDF)

## Acknowledgments

The authors gratefully acknowledge the help of Jan Kees van Amerongen for running the local computer cluster, and Bram van Dijk for proofreading the manuscript.

## Author Contributions

**Conceptualization:** Samuel H. A. von der Dunk, Paulien Hogeweg, Berend Snel.

**Investigation:** Samuel H. A. von der Dunk.

**Supervision:** Paulien Hogeweg, Berend Snel.

**Visualization:** Samuel H. A. von der Dunk.

**Writing – original draft:** Samuel H. A. von der Dunk.

**Writing – review & editing:** Paulien Hogeweg, Berend Snel.

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
