## [Decision Letter · Decision Letter 0]

27 Sep 2023

Dear Mr. Von Der Dunk,

Thank you very much for submitting your manuscript "Intracellular signaling in proto-eukaryotes evolves to alleviate regulatory conflicts of endosymbiosis" for consideration at PLOS Computational Biology.

As with all papers reviewed by the journal, your manuscript was reviewed by members of the editorial board and by several independent reviewers. In light of the reviews (below this email), we would like to invite the resubmission of a significantly-revised version that takes into account the reviewers' comments.

We cannot make any decision about publication until we have seen the revised manuscript and your response to the reviewers' comments. Your revised manuscript is also likely to be sent to reviewers for further evaluation.

Sincerely,

Attila Csikász-Nagy

Academic Editor

PLOS Computational Biology

Zhaolei Zhang

Section Editor

PLOS Computational Biology

Reviewer's Responses to Questions

**Comments to the Authors:**

Reviewer #1: Understanding the coevolution of host and endosymbionts is interesting and the modeling approach taken here is potentially powerful. However, there are a few things that greatly temper my enthusiasm for the paper and having left me wanting to like this paper more than I do.

Why only 10 runs? I can’t recall simulation-based analyses that rely on so few, and having only 10 and limits the ability to make generalizations. The few results means that broad generalizations are being based on (sometimes) only one or two outcomes. Does one (or two, or three) outcome out of 10 substantiate any generalization?

I assume the number of runs is so few because they are computationally demanding, but I have to assume, because the authors do not provide information on computational time (such information should be provided). Even if the simulations, as run, are computationally demanding I’m left wondering whether attempts were made to simplify / alter the parameters to run things faster. For example, it is common in population genetic simulations to increase the mutation rate and or strength of selection to explore some situations.

The Results section is frustrating at times. Stories are told based on results of individual replicates, then broad claims are made about the consequences/effects of certain phenomena. But these claims are often made without clear reference to support those claims (and with so few replicates I don’t know how the authors can be so certain). In other words, individual cases are being used to make broad claims (e.g. line 235), and the data supporting the claims are not always shown. Too many “For example…..” (e.g. line 212),

The Results discuss replicate Q4 fairly extensively and the authors point out that this is a highly successful symbiont. It is interesting if this is correct, although again, with only 10 replicates one doesn’t know if Q4 really represents the best strategy. Moreover, I’m left wondering if the Q4 strategy is so successful, why is it seen in only one replicate? Are others trapped on local optima? Is mutation limitation preventing the other replicates from achieving a similar outcome? I would expect answers to such questions from a simulation study ….. even if not definitive answers, I would expect the authors to provide some insight.

The authors seem focused on explaining the results of their simulations, but any ties back to what we see in the natural world are weak – even though the intro claims the simulations are motivated as a way to explore the early stages of eukaryotes. One example, is the paragraph starting on line 202, which starts with the sentence that “Host control creates selection for equal genome sizes…” and concludes with the sentence “Thus, cell-cycle synchronization promotes symmetry in genome size despite functional differentiation as observed before.” But this is not the pattern we see in eukaryotes? So there appears a mismatch between the lessons from the simulations and biological reality? If I’m correct, then what are we learning from this model?

Line 320. I find it puzzling that if the motivation is to understand the early stages of eukariogenesis the model is starting with identical host and symbiont genomes. The authors themselves acknowledge that the real world scenario would be better captured by having t=0 genomes of hosts and symbionts that are not identical? This adds to the general feel of the paper – that it is more interested in the results of the model than using the model to understand biology.

Less substantive & minor issues:

Why are simulation data only “available upon request”? Most journals have moved away from this (and it is usually not long before it becomes quite difficult to obtain such data). Are the files too large to put on a publicly available repository?

Line 32 “relative to virtually all known…..which had already adapted to a endosymbionts (i.e. mitochondria)”. If there are endosymbiosis that are “known” that don’t involve a host with a mitochondria, then what are they (I’d like to know, and I assume other readers would too)? If non are known, then the “virtually all known” is not sensical.

Line 40 “..with one notable exception …”, good that there is a reference for this exception, but tell us what the exception is – if it is notable, then tell us, don’t make us go look at another paper.

Figure 1 provides a lot of data on a single replicate, and two other replicates are shown in the supplement. By why only 3? Why not provide these summary figures for all 10 replicates?

Figure 1 – the star apparently marks “the final population”….but it is not the final time in the simulation?

Line 125, I’m not at all convinced that 10 replicates of a simulation allows one to “evaluate the generality”

Line 130 “the invention” of signaling? Perhaps origin? or evolution?

Line 165 “…the single outcome of eukariogenesis that appeared 1–2 billion years ago….” I thought the first eukaryotes were much older, and then estimate was much more precise than the billion year range. A reference would help.

The paper would benefit from more references. For example, line 233 a claim is made about bi-direction control resembling a sizer mechanism for cell size control. But I don’t know how sizer mechanisms work – a reference to a paper that explores the evolution of sizer mechanisms belongs here.

Line 314 … “this is exploited by evolution…” but what is the selective advantage of transferring household [housekeeping] genes from symbiont to host?

Figures, horizontal axis .. .what is AUT?

Figure 4 caption….are these results from simulations with more distant genomes at t=0? Data are shown for 7 replicates, but how do I know which replicate belongs to which data?

Line 381. The authors report that the results presented here contrast with their “previous study (2023)”…but my understanding is that paper used the same model that was used here? So why the differing results? Seems odd the authors don’t even try to explain this (if it is the same model, then perhaps having only 10 replicates?).

Line 425 “we use as a basis our previous model of obligate endosymbiosis and cell-cycle regulation (von der Dunk et al. (2023),” “as a basis” implies this is not the same model. If not, then how does this model differ? If the same model, then why “as a basis”

Iine 428. The model has 4 states, but only 2 are identified?

Page 2 supplement claims that “ implicit control is by far the most common evolved strategy in our model” BUT, what does this mean with regard to the biological relevance of this model?

Page 3 supplement “strings of gene products that target more binding

sites are more conserved than those of gene products with no or few regulatory targets (see

“Gene family analysis” in Appendix 1 of Von der Dunk et al., 2022).” This is consistent with pleiotropic constraints, something seen with many signaling / gene / protein networks, the lack of reference to this literature is a bit concerning.

Page 3 supplement “impact of leakage and transfer is largest when evolution is started with identical

host and symbiont genomes, i.e. similar to the situation in the experiments described in the

main text. When host and symbiont genomes start out identical, genes likely retain some

of their ancestral overlap, resulting in the strongest molecular interference.” This is interesting, but what data support this claim?

In general, the supplement has some interesting discussion, but the nearly complete lack of references (only two papers by these authors) and general paucity of data leave me now knowing if this is just overinterpretation of a few replicates of a single simulation model, or the claims are actually well supported.

Figure S5, please repeat the meaning of the diamonds, triangles, colors, etc.

Reviewer #2: This paper by von der Dunk uses a computational model to explore mechanisms by which the coordination of cell cycle control between a host and endosymbiont can emerge during evolution. The study builds on previous work by the team, in particular on a previous paper by the same author on the evolution of cell cycle control.

The work is highly original and clever. There is no-one else in the world attempting this kind of analysis. The findings are interesting and promise to shed new light on the mechanisms and evolution of cell cycle control in the context of endosymbionts.

Thus, the paper will interest to a wide audience of researchers interested in cell and evolutionary biology.

The approach is interesting and sophisticated. The authors use a multilevel approach. Selection acts on the ability of holobionts possessing stochastic regulatory gene networks controlling DNA replication and division timing to populate a landscape with different levels of nutrients. During the course of evolution, the model allows for changes to genome architecture, gene transfer, changes to genes, their localization, and enhancers/promoters in both host and symbiont genomes.

The paper also attempts to dissect the effects of changes in i) the localization of regulatory machinery within the holobiont, ii) product leakage between host and symbiont, iii) and gene transfer between host and symbiont on the evolution, of successful holobiont propagation.

This leads to the interesting finding that while leakage creates problems via interference, it is a pre-requisite for the evolution of coordination - perhaps because genomes learn to live with one another.

After the evolution of successful holobionts, successful individuals are studied alone, in competition, and the role of leakage was assessed during their evolution.

All in all, the scope of the work is impressive!

In summary I would love to see a revised paper published.

However, in my view much needs to be done for it to make sense to an uninitiated reader.

General comments:

1, The rules of the game

While one could imagine the model having been formulated in many different ways, the approach has been carefully thought through. The rules are sensible.

The title and abstract are appropriate.

However, the paper would benefit from more effort to explain the way the computational model is formulated. As it stands the paper is thin on details and requires readers to do a lot of work and sit with the previous paper Von der Dunk 2022 open while they read this paper.

In my view this is not good enough, the paper should stand on its own.

This is important because the assumptions of the model and the evolution affect the way one should think about the results.

Details should be explained in a simple way, ideally in a table, and discussed in the paper.

While I may not have fully gleaned all the assumptions (because they have not been made very clear), as I understand it, the following are some of the critical assumptions for the way the model works:

A. Mechanisms of cell cycle control:

The time taken for DNA replication to be completed depends on nutrient availability and genome size.

Thus, there is selection for holobionts that can delay division in a way that depends on nutrient conditions to avoid dividing with under-replicated DNA.

There is no way for organisms to monitor whether DNA replication is underway.

The only way they can measure DNA replication is by assessing whether specific genes in the genome (e.g. genes far from the origin of replication) are present in two copies.

They can also survive if they possess a cell division cycle that is longer than the time required for DNA replication.

Question: Am I correct in thinking that once DNA is replicated it is assumed that cells never undergo a second round of DNA replication?

Question: Is there is a corresponding change in dosage related to numbers of TF binding sites (i.e. competition for TF binding)?

B. Symbiosis:

Holobionts cannot live without symbionts. Symbionts use nutrients. This creates selection for reduced symbiont copy number.

Symbiont division is assumed to occur at the same time as the host. There is no way to evolve machinery to ensure their fair segregation (like that present in eukaryotes), symbionts are segregated randomly at division. As a result there is selection for increased symbiont number.

All symbionts are assumed to have the same genome.

C. Gene expression:

Gene expression is based on gene copy number. As a result, it also depends on symbiont copy number.

There is only one regulatory binding TF binding site upstream of each gene. This can only be functional or dead, there is no in-between state.

Proteins can be targeted to symbiont or host or both based upon the signal sequence they possess.

Proteins can leak between symbiont and host. This leakage represents a fixed proportion of total protein.

Question: if TFs are targeted to both host and symbiont does this reduce their expression level?

Question:

What does this mean in the 2022 paper: "Noise is introduced through sequence-specific binding affinities between gene products and binding sites (where affinity is a function of bitstring similarity);"

And I right in thinking that although expression is “stochastic”, genes that are expressed are expressed in precise numbers that depend on genome copies, and when present in sufficient numbers act on all target genes?

How much protein is required to have an effect on downstream targets following leakage?

D. Selection:

Please explain how population size affects selection in the model.

Please explain how competition experiments differ from clonal experiments.

Apologies if I should have understood this, but can the model select for organisms that grow slowly without making errors (persisters), or is the model formulated in such a way that those that grow fast always win out?

E. Solutions enabling survival:

Solutions include:

i). Both symbiont and host evolve a similar cell cycle that works independently in the two.

ii) The cell cycle time is longer than the DNA replication time in the host and symbiont.

iii) Symbiont genomes are small - since they only divide once per cell cycle this helps and doesnt lead to increased number. Increased numbers only arise from stochastic divisions.

iv) There is a checkpoint such that division cannot occur until gene dosage is high enough.

Are there any others?

It is likely that this summary of the key assumptions and outcomes of the model contains errors.

Hence the need for clarity in the paper!

2. Figures and Figure legends

While there is much to praise in the paper and while the results are fascinating, the Figures and Figure legends are very confusing.

The first Figure should explain the approach - Figure 5.

Figure 2 should come before Figure 1.

Its weird to jump in with one that seems “eukaryotic like”

For me, Figure 3 is incredibly hard to understand.

Figures should be clear and should be broken up into sections, e.g. a-d, so that the reader can follow what is where. They are too crowded.

They are more like slides in a powerpoint presentation than Figures in a paper.

Figure legends must explain what is in the Figure.

This is not the case. Much is left unexplained.

As it stands the Figure legends discuss conclusions.

This is the wrong place to do this.

3. Analogies.

A. The authors make assumptions about what eukaryotes like that appear at odds with eukaryotic cell biology.

In eukaryotic cells, mitochondrial DNA replication is uncoupled from host DNA replication. It occurs throughout the cycle. Mitochondria can also undergo multiple rounds of DNA replication each cycle.

In addition, mitochondrial scission is not coupled to coupled to mitosis.

Eukaryotic cells also possess mechanisms to ensure fair mitochondrial segregation.

Finally, mitochondria aren’t a metabolic burden, they AID host cell growth.

These facts should be explained in the paper!

B. The paper repeatedly compares results with cell size control models.

If the authors wish to do this, they need to explain the equivalence.

They also need to explain current thinking about cell size control, and this needs to reference the most recent papers on the subject.

Most eukaryotic and prokaryotic systems are currently thought to be adders.

The text should properly explain models of cell growth and division.

1. Adder (add a fixed amount of mass per cycle).

2. Timer (divide after fixed time has elapsed). This never applies in systems that need to respond to environment. Only in development.

3. Sizer: (divide at specific size). Ensures that division is impacted by growth.

A key test of the cell size control mechanism used in cells is the study of the impact of both size on homeostasis, and the response time following size perturbations.

If the authors are serious about making the comparison with cell size, they should implement this by asking how the cell cycle is changed in cells with increased or decreased numbers of symbionts.

C. Line 225: There is a statement about cell ageing that is confusing.

D. Cell cycle checkpoints should be defined.

4. Statements not supported by data.

There are several cases in the paper where statements are made without the data being presented.

This is not appropriate. All instances of this should be fixed.

5. Signalling;

There was no experiment to allow leakage and transfer but no “signalling".

Why not? Please explain.

Specific comments on the paper:

Line 28:

It is not true to say that the nucleus has control over the mitochondria.

Mitochondrial DNA replication is not subject to host control, but is dependent on holobiont mass accumulation.

While scission occurs from the outside in many cases mitochondria, LECA likely had FtsZ:

https://www.pnas.org/doi/full/10.1073/pnas.1421392112

Line 43:

This is a hypothesis (without supporting references) and should be presented as such.

Line 59: This is not a good summary of cell size control models.

See above.

Line 73: The text should;d describe the evolutionary experiment,

How was it done?

How many rounds?

How many individuals were analysed from each experiment?

Line 74: Before diving into Q4, I'd like a summary of the results.

Line 83: What do the authors mean by “selective forces are weak”?

Explain how population size and selection work in the model.

Figure 1 Legend not clear enough.

Needs to be divided up into sub Figures e.g. a-d.

In the final part of the Figure, t=0 is insensitive to leakage - why?

Line 117: Where is data to support this statement about it being “essential” by t = 0.05?

Line 118/119: What’s the evidence that leakage helps coordination?

Line 120-121: Please show evidence!

Line 138 statement is too strong

Line 150-154: What does competition test? How do they compete? Is this different to growth speed?

Figure 3: The Figure is incredibly hard to understand.

The assumption is that host control is similar to eukaryotes.

This is not the case (see above). Eukaryotes have gene transfer. And coordination.

Moreover, the output of mitochondria affects cell growth and success.

Line 197: Statement needs supporting data.

A good test of control would be to alter symbiont number and assay return to steady state.

This is how diagnostics of cell size models are tested.

Line 200: If its all selection then there is no control over symbiont number.

Is this the case?

Line 205: This isnt how it works in eukaryotes.

Why shouldn’t mitochondria replicate outside of DNA rep cycle of host, and have smaller genomes?

Line 220: Not clear how 2 fold change can be read out by stochastic system - please explain.

Line 225: This does not resemble ageing.

This resembles a normal cell cycle!!!

Line 234:”usually” is overstatement.

Assumptions of the model should be presented clearly:

Line 243: Why is the timing of emergence of a network important? What does this tell us?

All it means is that its hard to evolve, e.g. human intelligence.

Line 254: would be good to know how many of the solutions rely on quantitative gene expression before and after DNA replication.

Line 267: please show data supporting dormant state.

How was this studied?

What about other systems?

Line 320: I really like this experiment.

A modified Figure 5 should be included as Figure 1.

Supplement:

What does the following mean?

"we setup an evolution experiment with 12 different host–symbiont pairs"

"Each host–symbiont pair was evolved in two or three separate replicates to untangle the effect of speciﬁc pairings.”

Figure S1 - what is “Standard”?

In supplement it says:

"Thus, if we combine these results with those from the experiments described in the main text, implicit control is by far the most common evolved strategy in our model. Signaling only emerges when relatively primitive hosts and symbionts (i.e. which are not adapted to poor nutrient conditions) are exposed to product leakage from the start.”

The Supplementary text should explain exactly what the results are and how experiments were done.This is discussion not supplement.

It says this:

"The inability to evolve signaling might be an important reason why holobiont popu-lations remain smaller in the presence of product leakage in particular”

Please explain.

Reviewer #3: The manuscript by Hogeweg and co-workers seek to explain how cell-cycle synchronization evolved for a host-symbiont system. The system starts with an obligate mutualism (they cannot live without each other) but without any influence on each other’s multiplication. Influencing the cell cycle of the other cell(s) can come about by implicitly leaking some peptides or targeting signals toward the other cell. The results show that multiple paths toward efficient cell-cycle control exists.

I like the manuscript; it is well written, and the results are explained in detail. My criticisms involve the presentation, both of which can be easily remedied.

(1) Do not use PLOS strange Intro-Results-Discussion-Method structure, I never understood why use it for theoretical papers which cannot be understood without going through the methods first. So structure the paper the usual way: Intro-Methods-Results-Discussion.

(2) My usual quibbles with manuscript heavily building on another paper is that the method section does not stands on its own, and the reader has to go to the other paper to understand what is going on. I think repetition in the method section should not be a problem between papers. For example, I was unable to find the initial number of genes in the host and the symbiont. What is the probability of deletion or duplication of a gene? What is the difference between the host and the symbiont (symbiont being smaller I presume)?

In summary, the manuscript should be accepted for publication with the above edits.

Minor comments

Line95: there is a duplicate of “whereas”

L103. The subscript influx should not be italicized

L469 The subscript s should not be italicized

L474 this sentence reads bad, maybe the preposition “in” should be “of”

L477 The subscript T should not be italicized

**Have the authors made all data and (if applicable) computational code underlying the findings in their manuscript fully available?**

Reviewer #1: **No: **Results are "available upon request" ...not sure if that is OK.

Reviewer #2: **No: **Many statements arent supported by data

Reviewer #3: Yes

PLOS authors have the option to publish the peer review history of their article (what does this mean?). If published, this will include your full peer review and any attached files.

Reviewer #1: No

Reviewer #2: No

Reviewer #3: **Yes: **Ádám Kun
---

## [Decision Letter · Decision Letter 1]

21 Dec 2023

Dear Mr. Von Der Dunk,

Thank you very much for submitting your manuscript "Intracellular signaling in proto-eukaryotes evolves to alleviate regulatory conflicts of endosymbiosis" for consideration at PLOS Computational Biology. As with all papers reviewed by the journal, your manuscript was reviewed by members of the editorial board and by several independent reviewers. The reviewers appreciated the attention to an important topic. Based on the reviews, we are likely to accept this manuscript for publication, providing that you modify the manuscript according to the review recommendations.

Please add a paragraph describing the limitations of the approach and explain how computationally expensive to run simulations.

Sincerely,

Attila Csikász-Nagy

Academic Editor

PLOS Computational Biology

Zhaolei Zhang

Section Editor

PLOS Computational Biology

Reviewer's Responses to Questions

**Comments to the Authors:**

Reviewer #1: I reviewed a previous version of this manuscript. My view of this version remains largely unchanged: very interesting question but the work seems more concerned with understanding their model and understanding the behaviour of single replicates. I find the weakest aspect to be that this is a simulation paper yet only 10 simulations are run (sure, there are more than 10 for the entire paper – as the response letter points out), but there are only 10 that examine the evolution of the symbiosis, and the others follow up on some of those 10. I fundamentally don’t see how 10 replicates of a simulation enable one to really evaluate general principles. It is easy to develop very computationally intensive models, but they are not always particularly useful.

I realize others have a different perspective and of course they are not going to change their model. Given that, I do think this would be a better paper and better serve the community if there was a paragraph the described the limitations and caveats of the model.

A rather minor point, but if the authors justify only 10 replicates due to computational demands, then I’m puzzled as to why they don’t report how computationally demanding the model is.

Reviewer #2: I would like to congratulate the authors on their hard work in the revision.

The paper addresses all the reviewers comments in a very considered way.

The paper is also much clearer, and an easier read.

For these reasons, I would be happy to see the paper published in its current form.

Its a very original study that will interest a wide audience of evolutionary and cell biologists.

2 very minor corrections:

I would suggest modifying the following lines to:

Line 74: "growth and division cycles relative"

Line 236: “establish that product leakage drives the evolution”"

Reviewer #3: My comments were fully addressed, and thus I’m happy to recommend acceptance of this manuscript.

I think that the results section is a bit too long. I do not know how the shorten it, so it is not a demand on my part.

**Have the authors made all data and (if applicable) computational code underlying the findings in their manuscript fully available?**

Reviewer #1: None

Reviewer #2: Yes

Reviewer #3: Yes

PLOS authors have the option to publish the peer review history of their article (what does this mean?). If published, this will include your full peer review and any attached files.

Reviewer #1: No

Reviewer #2: No

Reviewer #3: **Yes: **Ádám Kun

Figure Files:

Data Requirements:

Reproducibility:

References:

---

## [Editor Report · Decision Letter 2]

24 Jan 2024

Dear Mr. Von Der Dunk,

We are pleased to inform you that your manuscript 'Intracellular signaling in proto-eukaryotes evolves to alleviate regulatory conflicts of endosymbiosis' has been provisionally accepted for publication in PLOS Computational Biology.

Best regards,

Attila Csikász-Nagy

Academic Editor

PLOS Computational Biology

Zhaolei Zhang

Section Editor

PLOS Computational Biology

---

## [Editor Report · Acceptance letter]

6 Feb 2024

PCOMPBIOL-D-23-01322R2 

Intracellular signaling in proto-eukaryotes evolves to alleviate regulatory conflicts of endosymbiosis

Dear Dr von Der Dunk,

I am pleased to inform you that your manuscript has been formally accepted for publication in PLOS Computational Biology. Your manuscript is now with our production department and you will be notified of the publication date in due course.

With kind regards,

Anita Estes
